# YAP controls retinal stem cell DNA replication timing and genomic stability

**Pauline Cabochette[1†], Guillermo Vega-Lopez[1†], Juliette Bitard[1,2], Karine Parain[3], Romain Chemouny[1], Christel Masson[1,2], Caroline Borday[1], Marie Hedderich[3], Kristine A Henningfeld[3], Morgane Locker[1], Odile Bronchain[1]\*, Muriel Perron[1,2,4]\***

[1]Paris-Saclay Institute of Neuroscience, CNRS, Université Paris Sud, Orsay, France; [2]Centre d'Etude et de Recherche Thérapeutique en Ophtalmologie, Retina France, Orsay, France; [3]Center for Nanoscale Microscopy and Molecular Physiology of the Brain, Institute of Developmental Biochemistry, University of Goettingen, Goettingen, Germany; [4]Jules Stein Eye Institute, University of California Los Angeles, Los Angeles, United States

**Abstract** The adult frog retina retains a reservoir of active neural stem cells that contribute to continuous eye growth throughout life. We found that *Yap*, a downstream effector of the Hippo pathway, is specifically expressed in these stem cells. *Yap* knock-down leads to an accelerated S-phase and an abnormal progression of DNA replication, a phenotype likely mediated by upregulation of *c-Myc*. This is associated with an increased occurrence of DNA damage and eventually p53-p21 pathway-mediated cell death. Finally, we identified PKNOX1, a transcription factor involved in the maintenance of genomic stability, as a functional and physical interactant of YAP. Altogether, we propose that YAP is required in adult retinal stem cells to regulate the temporal firing of replication origins and quality control of replicated DNA. Our data reinforce the view that specific mechanisms dedicated to S-phase control are at work in stem cells to protect them from genomic instability.

\*For correspondence: odile.
bronchain@u-psud.fr (OB);
muriel.perron@u-psud.fr (MP)

†These authors contributed
equally to this work

**Competing interests:** The
authors declare that no
competing interests exist.

**Reviewing editor:** Helen McNeill,
The Samuel Lunenfeld Research
Institute, Canada

## Introduction

Adult stem cell maintenance is required to sustain long-term preservation of tissue homeostasis. In the fish or amphibian retina, a continuously proliferating peripheral domain called ciliary marginal zone (CMZ) (*Wetts et al., 1989*; *Perron et al., 1998*) was recently formally demonstrated to contain genuine multipotent and self-renewing neural stem cells (*Centanin et al., 2011*). The CMZ not only ensures cell replacement, but also contributes to life-long growth of the eye through the permanent generation of all retinal cell types. The CMZ thus represents an ideal model for dissecting molecular cues underlying retinal stem cell properties in vivo. Such knowledge is essential for the development of innovative therapeutic strategies based on the mobilization and targeted activation of endogenous neural stem cells for tissue repair.

The Hippo pathway effector yes-associated protein (YAP) was identified as a major regulator of organ growth through its actions on embryonic precursor cells (*Lian et al., 2010*; *Ramos and Camargo, 2012*). YAP function in adult stem cells, however, remains unclear. For instance, *Yap* overexpression increases self-renewal of airway basal stem cells (*Zhao et al., 2014*). In contrast, it surprisingly leads to a loss of intestinal stem cells (*Barry et al., 2013*), while being seemingly neutral regarding the quantity and function of hematopoietic stem cells (*Jansson and Larsson, 2012*). Inactivation studies further suggested that YAP is largely dispensable in a physiological context for the homeostasis of several adult organs (*Cai et al., 2010*; *Azzolin et al., 2014*; *Chen et al., 2014*; *Zhang et al., 2014*), although this might reflect in some cases functional redundancy with the other Hippo effector TAZ

**eLife digest** In animals, stem cells divide to produce the new cells needed to grow and renew tissues and organs. Understanding the biology of these cells is of the utmost importance for developing new treatments for a wide range of human diseases, including neurodegenerative diseases and cancer. Before a stem cell divides, it copies its DNA and the two sets of genetic instructions are then separated so that the two daughter cells both have a complete set. This process needs to be as accurate as possible because any errors would result in incorrect genetic information being passed on to the daughter cells.

Stem cells in the light-sensitive part of the eye—called the retina—divide to produce the cells that detect light and relay visual information to the brain. In many animals, these stem cells stop dividing soon after birth and the retina stops growing. However, the stem cells in frogs and fish continue to divide throughout the life of the animal, which enables the eye to keep on growing.

A protein called YAP regulates the growth of organs in animal embryos, but it is not clear what role this protein plays in stem cells, particularly after birth. To address this question, Cabochette et al. studied YAP in the retina of frog tadpoles. The experiments show that YAP is produced in the stem cells of the retina after birth and is required for the retina to continue to grow.

Cabochette et al. used tools called 'photo-cleavable Morpholinos' to alter the production of YAP in adult stem cells. The cells that produced less YAP copied their DNA more quickly and more of their DNA became damaged, which eventually led to the death of these cells. Further experiments revealed that YAP interacts with a protein called PKNOX1, which is involved in maintaining the integrity of DNA.

Cabochette et al.'s findings provide the first insights into how YAP works in the stem cells of the retina and demonstrate that it plays a crucial role in regulating when DNA is copied. A future challenge is to find out whether YAP plays a similar role in the stem cells of other organs in adult animals.

(*Imajo et al., 2015*). YAP is implicated in tissue regeneration but its effects are controversial (*Cai et al., 2010*; *Barry et al., 2013*). Thus, the role of YAP in vertebrate adult stem cells may likely be context-dependent and clearly deserves further investigation. Since its function in adult neural stem cells is presently unknown, we took advantage of the *Xenopus* CMZ model system and investigated whether *Yap* is involved in the maintenance of an active pool of retinal stem cells in the continuously growing post-embryonic frog eye. Although YAP gain of function led quite expectedly to CMZ cell overproliferation, the loss of function analysis revealed a more complex phenotype. Indeed, we found that stem cells were still present but exhibited aberrant cell cycle progression. In particular, DNA replication timing was found to be altered leading to a dramatic S-phase shortening. This correlates with increased DNA damage and eventually cell death. We also found that YAP functionally and physically interacts with PKNOX1, a transcription factor required to maintain genomic stability (*Iotti et al., 2011*).

## Results

### *Yap* is expressed in slow dividing stem cells of the post-embryonic retina

In situ hybridization at the optic vesicle stage revealed prominent *Yap* expression in the presumptive retinal pigmented epithelium (RPE) and in the neural retina/RPE border (*Figure 1—figure supplement 1A*), a region we previously proposed to be the presumptive adult stem cell niche (*El Yakoubi et al., 2012*). In line with this, we found that in the post-embryonic retina, *Yap* is expressed in the most peripheral stem cell-containing region of the CMZ (*Figure 1A,B*). We also performed immunostaining using an antibody whose specificity was assessed in a loss of function context, that is, in tadpoles injected with *Yap* Morpholinos (*Yap*-MO; *Figure 1—figure supplement 2*; see also *Figure 2—figure supplement 1* for efficiency and specificity evaluation of *Yap*-MO). YAP protein was detected in stem cells located at the tip of the CMZ, mainly in the cytoplasm, although some signal could be observed as well in the nuclei of these cells. Of note, we also found YAP labeling in Müller glial cells (*Figure 1C*). To delineate more precisely the *Yap* expression domain, we co-labeled *Yap* and proliferative cells (*Figure 1D*). A short EdU pulse was performed allowing slow dividing stem cells to be distinguished from fast

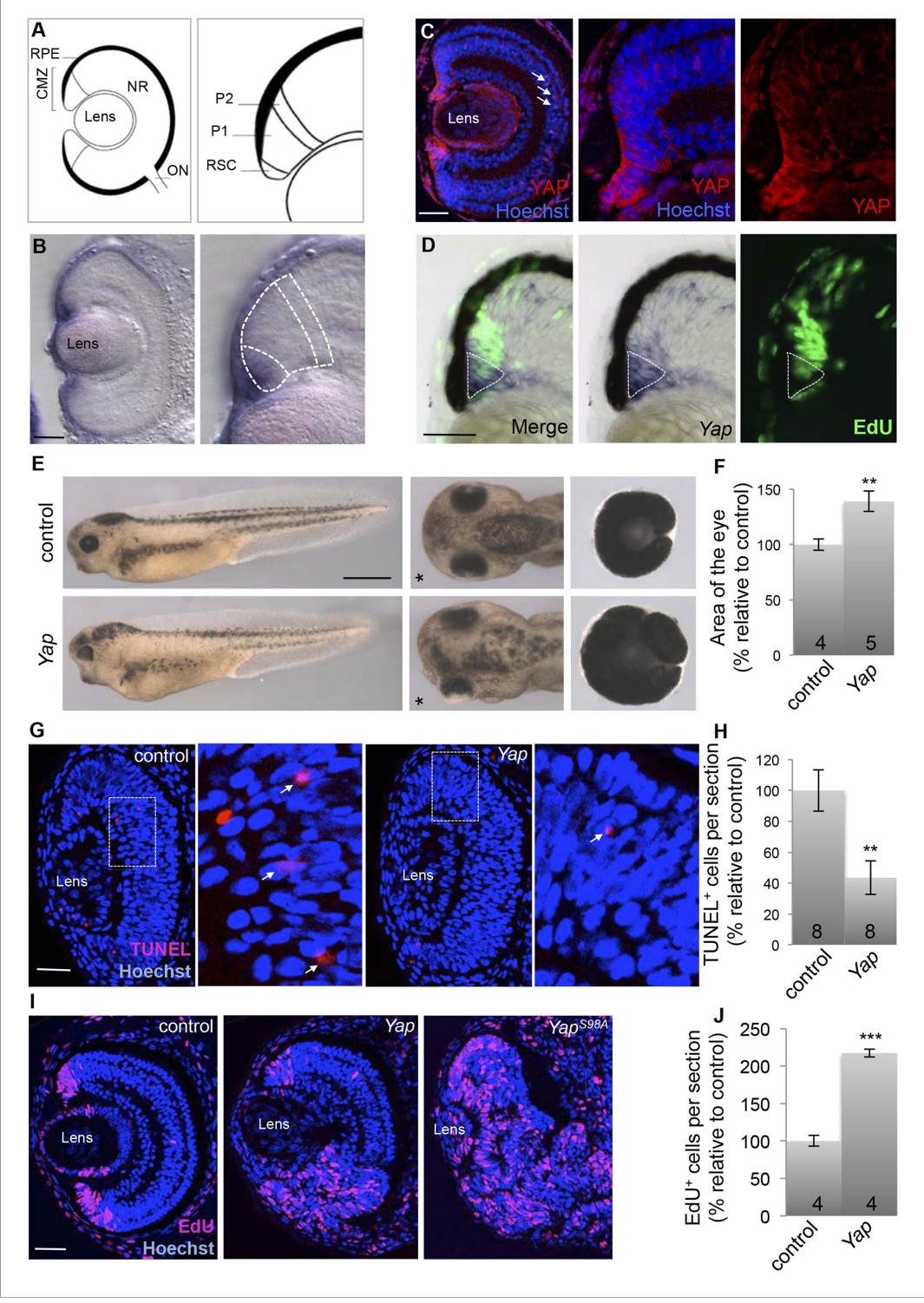

**Figure 1**. *Yap* overexpression expands the proliferating cell population in the post-embryonic retina. (**A**) Schematic transversal section of a Xenopus tadpole retina (RPE: retinal pigment epithelium; NR: neural retina; ON: optic nerve). Within the CMZ (right panel), retinal stem cells (RSC) reside in the most peripheral margin while actively dividing progenitors (P1) and their post-mitotic progeny (P2) are localized more centrally. (**B**) In situ hybridization analysis of *Yap* expression on stage 40 retinal sections. The image on the right is a higher magnification of the CMZ (dashed lines represent the different zones as in a). (**C**) Immunostaining with anti-YAP antibody on stage 42 retinal sections. YAP labeling is detected in the CMZ as well as in Müller glial cells (arrows). Images on the right are higher

*Figure 1. continued on next page*

*Figure 1. Continued*

magnifications of the CMZ. (**D**) EdU labeling (3-hr pulse) following in situ hybridization with a *Yap* probe (dotted line) on stage 40 retinal sections. (**E**) Lateral views (left panels), head dorsal views (middle panels) and dissected eyes (right panels) of stage 40 tadpoles following two-cell stage microinjection of *GFP* mRNA as a lineage tracer with either *ß-gal* (control) or *Yap* mRNA. The asterisk indicates the injected side. (**F**) Quantification of dissected eye area. (**G–J**) TUNEL (**G**, **H**; stage 33/34) or EdU incorporation (**I**, **J**; 3-hr pulse at stage 40) assays analyzed on retinal sections from tadpoles injected as in (**E**). Arrows point to TUNEL-positive cells in higher magnifications of the area delineated with dotted line (**G**). The number of analyzed retinas is indicated in each bar. Data are represented as mean ± SEM. Scale bar = 1 mm in (**E**) and 40 μm in other panels.

The following figure supplements are available for figure 1:

**Figure supplement 1**. *Yap, Taz* and *Tead* expression.

**Figure supplement 2**. Validation of YAP antibody specificity.

**Figure supplement 3**. *Yap^{ΔTBS}* does not promote CMZ cell proliferation.

proliferating transit amplifying progenitors in the CMZ (*Xue and Harris, 2011*). *Yap* staining was found to be prominent in EdU-negative stem cells and in the most peripheral EdU-positive cells (young progenitors). The staining then waned in more central older progenitor cells. Of note, in contrast to *Yap*, *Taz* is faintly expressed in the post-embryonic retina and only a weak and diffuse signal could be detected in the CMZ (*Figure 1—figure supplement 1B*).

Finally, as YAP acts as a co-transcriptional activator, we wondered whether its classical partners of the TEAD family were also expressed in the CMZ. We found consistent labeling of both *Tead1* and *Tead2* in the periphery of the CMZ where *Yap* is expressed (*Figure 1—figure supplement 1C*).

## *Yap* overexpression promotes post-embryonic eye overgrowth

To investigate YAP function in the post-embryonic retina, we first undertook a gain of function approach by the means of mRNA injection at the two-cell stage. *Yap*-overexpressing tadpoles displayed eye overgrowth on the injected side (*Figure 1E,F*). This phenotype prompted us to analyze the impact of YAP on both cell death and proliferation. We found that *Yap* overexpression results in both a decreased number of TUNEL-positive cells (*Figure 1G,H*) and a dramatic expansion of the EdU-positive cell population (*Figure 1I,J*). The overproliferative phenotype was strongly exacerbated upon overexpression of a *Yap* mutant construct where Ser-98 was replaced by an alanine (*Yap^{S98A}*) (*Figure 1I*). This residue (Ser-127 in mammalian YAP) is a conserved Lats phosphorylation site that has been shown to mediate the growth-suppressive output of the Hippo signaling cascade in vivo (*Zhao et al., 2007*). In contrast, overexpression of a truncated construct lacking the TEAD binding site (*Yap^{ΔTBS}*) was unable to trigger enhanced proliferation in the CMZ, suggesting that the overproliferative phenotype requires interaction with a TEAD protein (*Figure 1—figure supplement 3*). Together, these data reveal that *Yap*-dependent retinal overgrowth is likely caused by enhanced cell survival and cell proliferation.

## *Yap* knockdown reduces post-embryonic eye size

We next sought to determine whether *Yap* is essential for post-embryonic retinal growth by knocking it down using *Yap*-MO. The Morpholino concentration was chosen to efficiently decrease YAP quantity (as inferred from Western-blot analysis; *Figure 2—figure supplement 1A*), while avoiding previously described early embryonic defects (*Gee et al., 2011*). In these conditions, morphant tadpoles developed but exhibited significantly reduced eye size compared to controls (*Figure 2A,B*). Importantly, this phenotype was restored upon co-injection of *Yap*-MO with non-targetable *Yap* mRNAs, demonstrating specificity (*Figure 2—figure supplement 1B,C*). To exclude potential growth impairment at the level of the whole organism and assess the tissue autonomy of eye size defects, we performed optic vesicle isotopic and isochronic graft experiments (*Figure 2C*). When the optic vesicle of a control tailbud was transplanted into an enucleated morphant embryo, it nevertheless reached a normal size. In contrast, *Yap*-MO optic vesicles grafted in a control host generated smaller eyes, which is in accordance with *Yap* knockdown effects being eye autonomous.

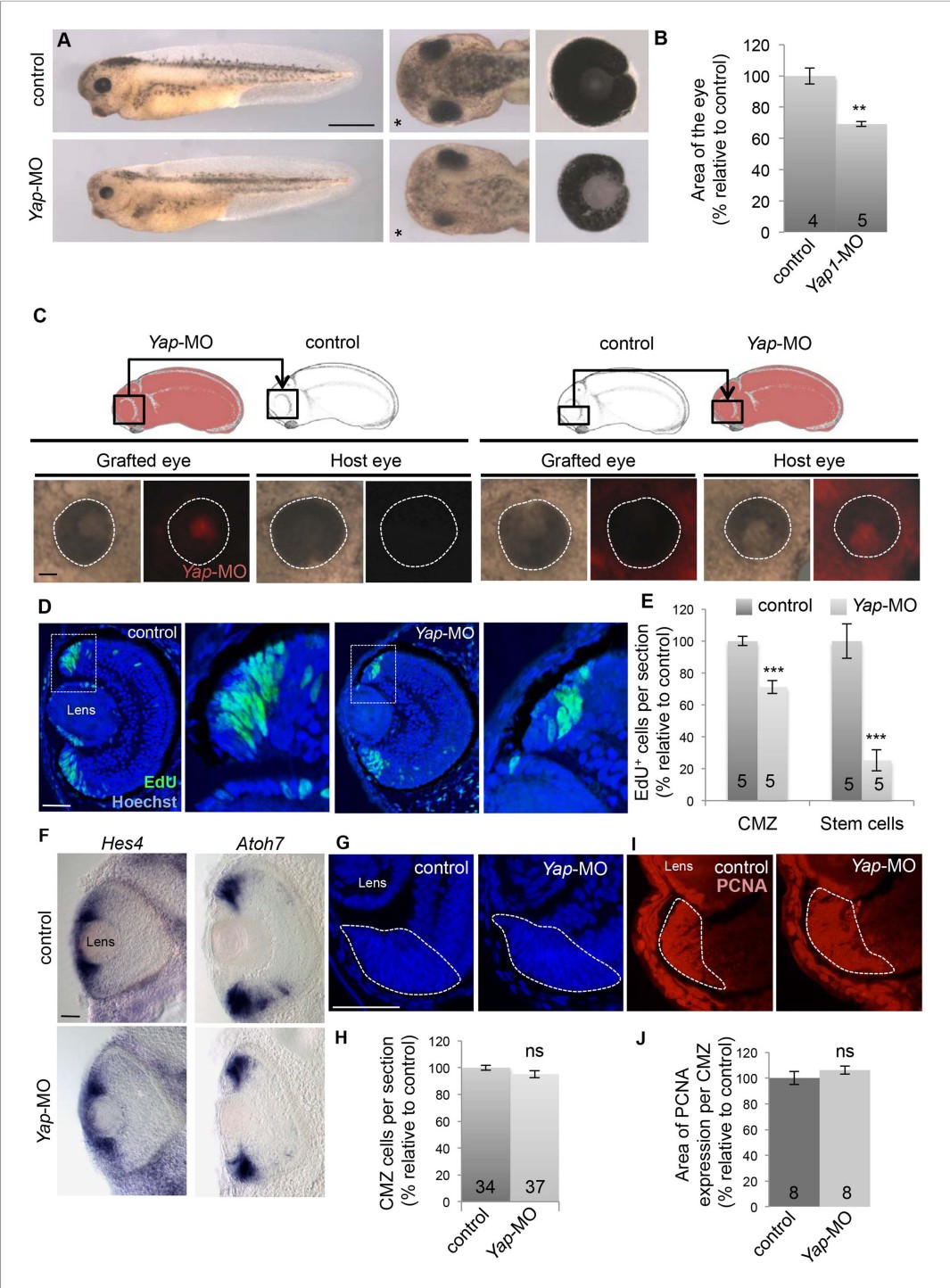

**Figure 2**. *Yap* knockdown decreases eye size and EdU incorporation in the post-embryonic retina. (**A**) Lateral views (left panels), head dorsal views (middle panels) and dissected eyes (right panels) of stage 40 tadpoles following two-cell stage microinjection of *Yap*-5-mismatch-MO (control) or *Yap*-MO. The asterisk indicates the injected side. (**B**) Quantification of dissected eye area. (**C**) Eyes of stage 40 tadpoles following optic vesicle transplantation at stage 25 as shown in the schematic. Dotted lines delineate the eye circumference. (**D**) EdU incorporation assay (6-hr pulse) analyzed on retinal sections from stage 40 tadpoles injected as in (**A**). Higher magnifications of the CMZ (delineated by dotted lines) are shown for each condition. (**E**) Quantification of EdU-positive cells within the whole CMZ or within the most peripheral stem cell compartment. (**F**) In situ hybridization analysis of *Hes4* (retinal stem cell marker; *El Yakoubi et al., 2012*) and *Atoh7* (progenitor cell marker; *Kanekar et al., 1997*) expression on stage 40 retinal

*Figure 2. Continued*

sections from tadpoles injected as in (**A**). (**G–J**) Hoechst staining and PCNA immunolabeling on stage 40 retinal sections from tadpoles injected as in (**A**). The CMZ is delineated with dotted lines. The number of analyzed retinas per condition is indicated in each bar. Data are represented as mean ± SEM. Scale bar = 1 mm in (**A**) and 40 µm in other panels.

The following figure supplement is available for figure 2:

**Figure supplement 1**. Validation of *Yap*-MO efficiency and specificity.

Finally, to address whether the reduced eye size was due to abnormal embryonic morphogenesis or to post-embryonic growth defects, we adapted in *Xenopus* the use of photo-cleavable Morpholinos (photo-MO). This technology, previously set up in zebrafish (*Tallafuss et al., 2012*), allows for an inducible or reversible gene knockdown through UV-induced cleavage of either sense or antisense photo-MOs (*Figure 3A*). We found that restoring YAP function at late embryogenesis, following knockdown during development, leads to tadpoles with normal sized eyes (*Figure 3B–D*). This supports the hypothesis that the *Yap*-MO phenotype is not the indirect consequence of developmental morphogenetic defects. In line with this, conditional *Yap* knockdown starting at late retinogenesis was found to be sufficient to trigger a small eye phenotype (*Figure 3B,E,F*). Together, these data point to a specific role for YAP in the homeostatic control of post-embryonic retinal growth.

## *Yap* knockdown impedes retinal stem cell proliferative activity

To investigate the cause of eye size reduction in *Yap* morphant tadpoles, we determined the level of proliferation within the CMZ. *Yap*-MO-injected tadpoles harbored a significantly decreased number of EdU$^+$ cells compared to a control situation (*Figure 2D,E*). The same was true when *Yap* knockdown was conditionally induced from late embryogenesis (*Figure 3—figure supplement 1*). The difference in EdU-labeling between control and *Yap* morphant tadpoles was even more pronounced at the tip of the CMZ where stem cells reside and *Yap* expression is the strongest (*Figure 2E*). We thus reasoned that *Yap* knockdown might decrease the number of proliferative cells in the CMZ as a consequence of stem cell depletion. Using in situ hybridization, we examined the expression of several retinal stem and progenitor cell markers (*Figure 2F* and data not shown). Surprisingly, stainings observed in *Yap*-MO-injected tadpoles were similar to control ones, indicating that both stem and progenitor cell populations were still present. Accordingly, neither the total number of cells within the CMZ nor the size of the proliferative cell population (PCNA labeled) was significantly affected (*Figure 2G–J*). Together, these data indicate that *Yap* knockdown does not induce stem/progenitor cell exhaustion but rather alters the relative proportion of time these cells spend in S-phase of the cell cycle.

## *Yap* loss of function affects cell cycle progression within the CMZ

The observed phenotype suggests that cell cycle kinetics of retinal stem/progenitor cells is perturbed in morphant tadpoles. As a first global approach to test this hypothesis, we evaluated the mitotic index in the whole post-embryonic CMZ using the mitotic marker phospho-histone H3 (PH3; *Figure 4A,B*). We found that compared to the control, *Yap* knockdown results in a significantly lower percentage of mitotic cells per section, which is suggestive of a longer total cell cycle length. To further investigate cell cycle progression defects at the level of the whole CMZ, we then measured G2 length using the percentage of labeled mitosis (PLM) paradigm (*Quastler and Sherman, 1959*; *Cai et al., 1997*; *Locker et al., 2006*) (*Figure 4C–E*). As expected from the asynchrony among retinal cells in the CMZ, the percentage of PH3/EdU-labeled cells increased sigmoidally with increasing EdU exposure times, before reaching a plateau (*Figure 4E*). Noticeably, the PLM was consistently lower at each time point in *Yap* morphant retinas compared to control ones, indicating a delayed S- to M-phase progression. We estimated that the mean G2 length (T$_{G2}$; see 'Materials and methods' for details) doubled in *Yap* morphant retinas compared to the control (from approximately 2.5 to 4.5 hr). It thus appears that *Yap* knockdown results in perturbed cell cycle kinetics within the CMZ.

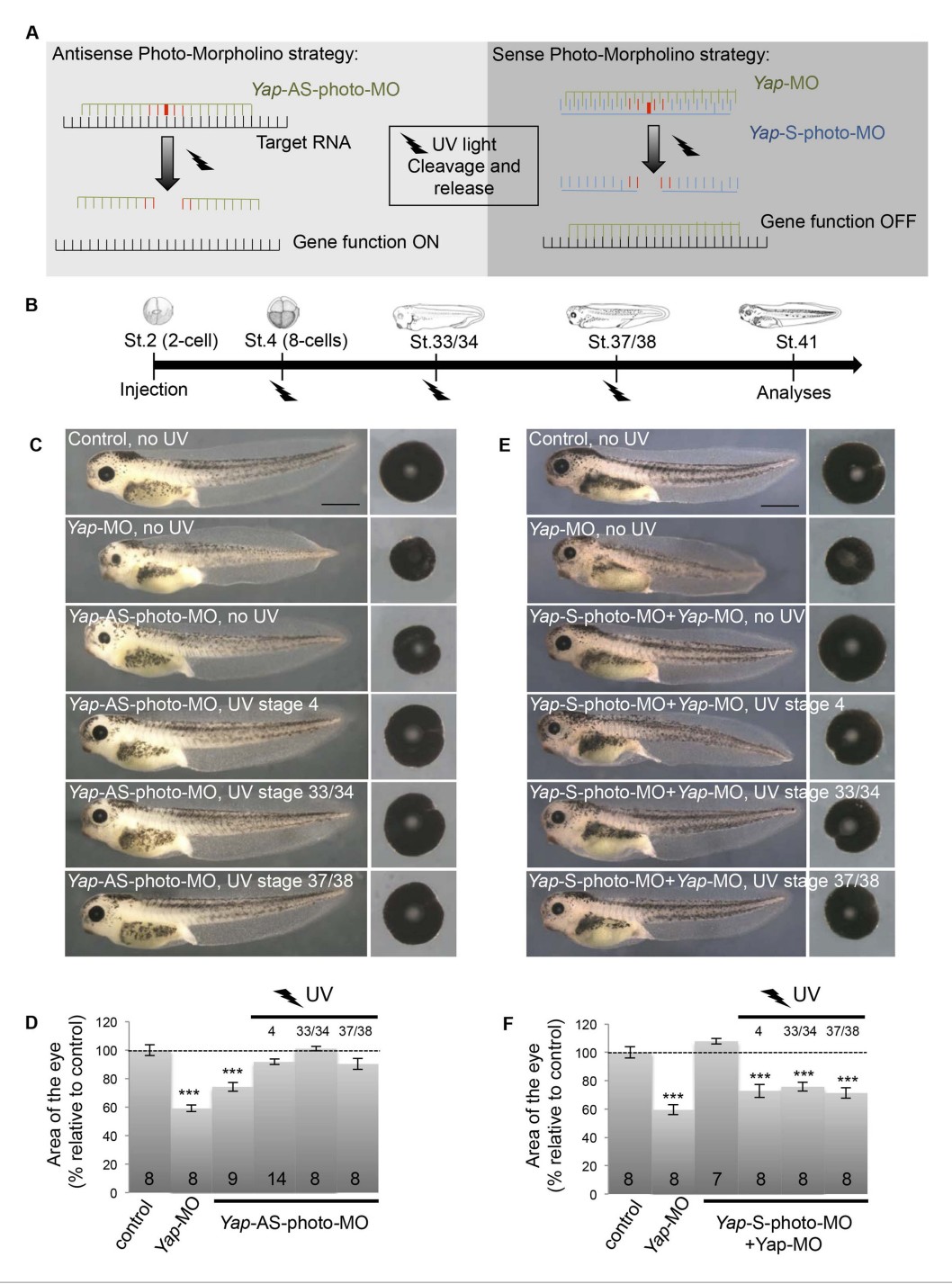

**Figure 3**. Conditional *Yap* knockdown in the retina. (**A**) Principle of reversible and inducible gene knockdown using photo-Morpholinos (photo-MO). Photo-MO contains a photo-sensitive subunit cleaved by 365 nm light. *Yap*-AS-photo-MO is degraded upon UV light exposure and its translation blocking activity is thus interrupted. Unmodified *Yap*-MO is rendered inactive by binding to *Yap*-S-photo-MO. It therefore cannot bind its mRNA target until light-induced cleavage of the sense MO. (**B**) Diagram of the experimental design. Embryos are microinjected with MO at the two-cell stage, subjected to UV exposure at different developmental stages as indicated (black flashes) and then sacrificed for analyses. (**C–F**) Analysis of reversible (**C**, **D**) and inducible (**E**, **F**) *Yap* knockdown. (**C**, **E**) Lateral views and dissected eyes of stage 41 tadpoles following two-cell stage microinjection of the indicated MO (see table in B). (**D**, **F**) Quantification of dissected eye area. The stage at which UV exposure was performed is indicated above each bar. The *Yap*-AS-photo-MO (without any UV exposure) shows the same efficiency

*Figure 3. continued on next page*

*Figure 3. Continued*

as the *Yap*-MO in reducing eye size. It is efficiently cleaved by UV light since exposure right after injection (stage 4) leads to a wild type phenotype. Restoring *Yap* function from stage 33/34 or even from stage 37/38 onwards leads to normal eye sized embryos, demonstrating that restricting *Yap* knockdown to embryogenesis is not sufficient to affect tadpole eye growth. The *Yap*-S-photo-MO efficiently blocks *Yap*-MO since their co-injection does not affect eye size. It is efficiently cleaved by UV light since exposure right after co-injection (stage 4) leads to a small eye phenotype, as observed in *Yap*-MO-injected embryos. Conditional *Yap* knockdown by light exposure from stage 33/34 or even from stage 37/38 onwards is sufficient to reduce eye size, suggesting that *Yap* is required at post-embryonic stages to maintain CMZ-dependent eye growth. Of note, in our experimental conditions, UV light exposure does not generate any significant effects on eye size (data not shown). The number of analyzed tadpoles is indicated within each bar. Scale bar = 1 mm.

The following figure supplement is available for figure 3:

**Figure supplement 1**. Inducible *Yap* knockdown at post-embryonic stages reduces EdU incorporation in the CMZ.

## *Yap* loss of function affects S-phase duration in retinal stem cells

In order to specifically measure total cycle length ($T_C$) of retinal stem cells, we next turned to a cumulative labeling assay (*Nowakowski et al., 1989*), a well-established technique that also allows evaluating S-phase ($T_S$) length (see 'Materials and methods' for details). As shown in *Figure 4F*, the labeling index in the linear part of the curve was consistently lower in *Yap* morphant retinal stem cells compared to control ones. Calculation of $T_C$ confirmed the hypothesis of extended cell cycle duration following *Yap*-Mo injection. Surprisingly, it also revealed a dramatic reduction of S-phase length in morphant cells (*Figure 4G*).

Such unexpected S-phase shortening prompted us to further investigate the underlying molecular mechanisms. In eukaryotes, origins of replication are activated throughout the S-phase in a temporally controlled manner such that some origins fire early and others fire late. The *c-Myc* proto-oncogene has been shown to accelerate S-phase by inducing premature origin firing initiation and increasing origin density (*Robinson et al., 2009*; *Srinivasan et al., 2013*). In situ hybridization and qPCR analyses revealed an upregulation and expansion of *c-Myc* expression in the CMZ of the *Yap* morphant retina (*Figure 5A–C*). This may account, at least in part, for the S-phase length shortening. To strengthen this hypothesis, we examined EdU-labeled replication foci at the tip of the CMZ. Their abundance and distribution, as classically observed in synchronized cultured cells, are indeed known to differ between early (numerous small foci located throughout the nucleus) and mid/late S-phase (limited number of large foci; *Figure 5D*) (*van Dierendonck et al., 1989*; *Koberna et al., 2005*). In the control situation, we found both early and late patterns within the stem cell population of the CMZ. In contrast, the proportion of cells in mid/late S-phase was dramatically reduced in *Yap* morphant cells (*Figure 5E,F*). Altogether, these data highlight that loss of YAP function in retinal stem cells alters their temporal program of DNA replication and points to *c-Myc* as a potential actor in this process. Interestingly, a similar phenotype (decrease in late replication patterns and *c-Myc* up-regulation) was also observed in the ventricular zone of the neural tube where *Yap* is expressed, suggesting that YAP function in S-phase progression may also hold true in other neural precursor cells (*Figure 6A–D*).

## *Yap* loss of function induces DNA damage

DNA replication stress results in DNA damage and consequent genomic instability (*Zeman and Cimprich, 2014*). We thus examined the expression of phosphorylated histone H2AX (γ-H2AX), the most sensitive marker for DNA double-strand breaks (*Rogakou et al., 1998*). The number of γ-H2AX-positive cells was significantly increased in *Yap*-MO-injected CMZ compared to controls (*Figure 7A,B*). Since extensive DNA damage may trigger apoptosis, we next turned to a TUNEL assay and found that cell death was indeed severely increased in morphant retinas (*Figure 7C,D*). Surprisingly, the majority of apoptotic cells was found at 'the exit' of the CMZ close to the neural retina, and not in its most peripheral part where *Yap* is expressed. This strongly suggests that apoptosis occurs as a secondary consequence in late progenitor cells generated from stem cells that experienced YAP function inhibition.

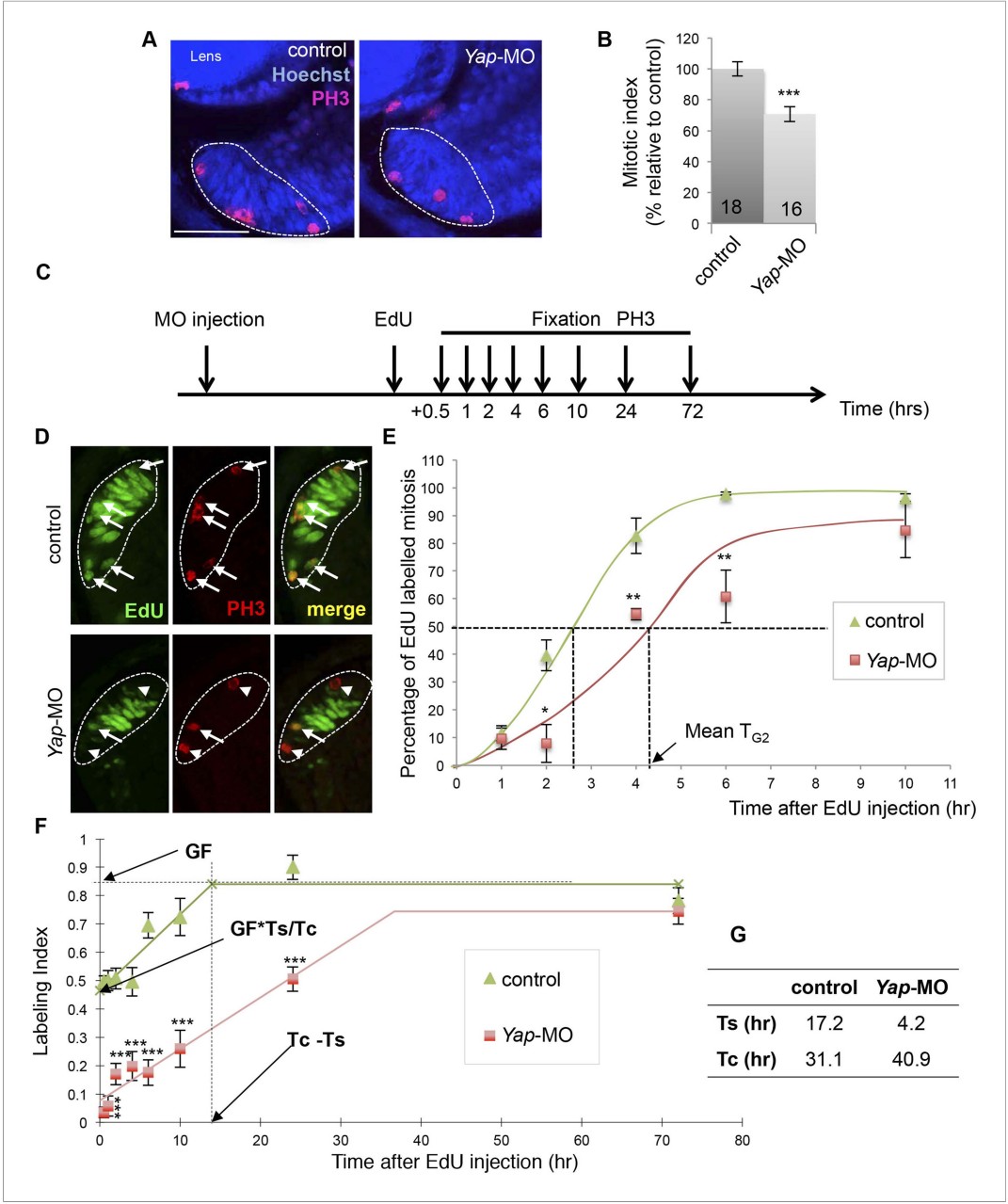

**Figure 4**. *Yap* loss of function slows down cell cycle kinetics of retinal stem cells. (**A**) PH3 immunolabeling on retinal sections from stage 40 tadpoles following two-cell stage microinjection of *Yap*-5-mismatch-MO (control) or *Yap*-MO. The CMZ is delineated with dotted lines. (**B**) Quantification of the mitotic index within the CMZ. The number of analyzed retinas per condition is indicated in each bar. (**C**) Outline of the PLM and EdU cumulative labeling experiments: tadpoles injected as in (**A**) were fixed at different time points following EdU injection at stage 39. EdU and PH3 labeling was then analyzed on retinal sections. (**D**) Retinal sections stained for both PH3 and EdU. The CMZ is delineated with dotted lines. Arrows and arrowheads point to PH3$^+$/EdU$^+$ and PH3$^+$/EdU$^-$ cells respectively. (**E**) Quantification of the PLM within the whole CMZ. $T_{G2}$: G2-phase duration. (**F**) Quantification of the EdU labeling index within retinal stem cells along with increasing EdU exposure times. GF: growth fraction; $T_C$: total cell cycle; $T_S$: S-phase. (**G**) Estimation of $T_C$ and $T_S$. Data are represented as mean ± SEM. Scale bar = 40 µm.

As stated above, the number of CMZ cells is not significantly changed in *Yap* morphant tadpoles and therefore retinal growth impairment cannot be simply explained by a depletion of the stem/progenitor pool. To foresee whether the increased cell death at the 'exit' of the CMZ could

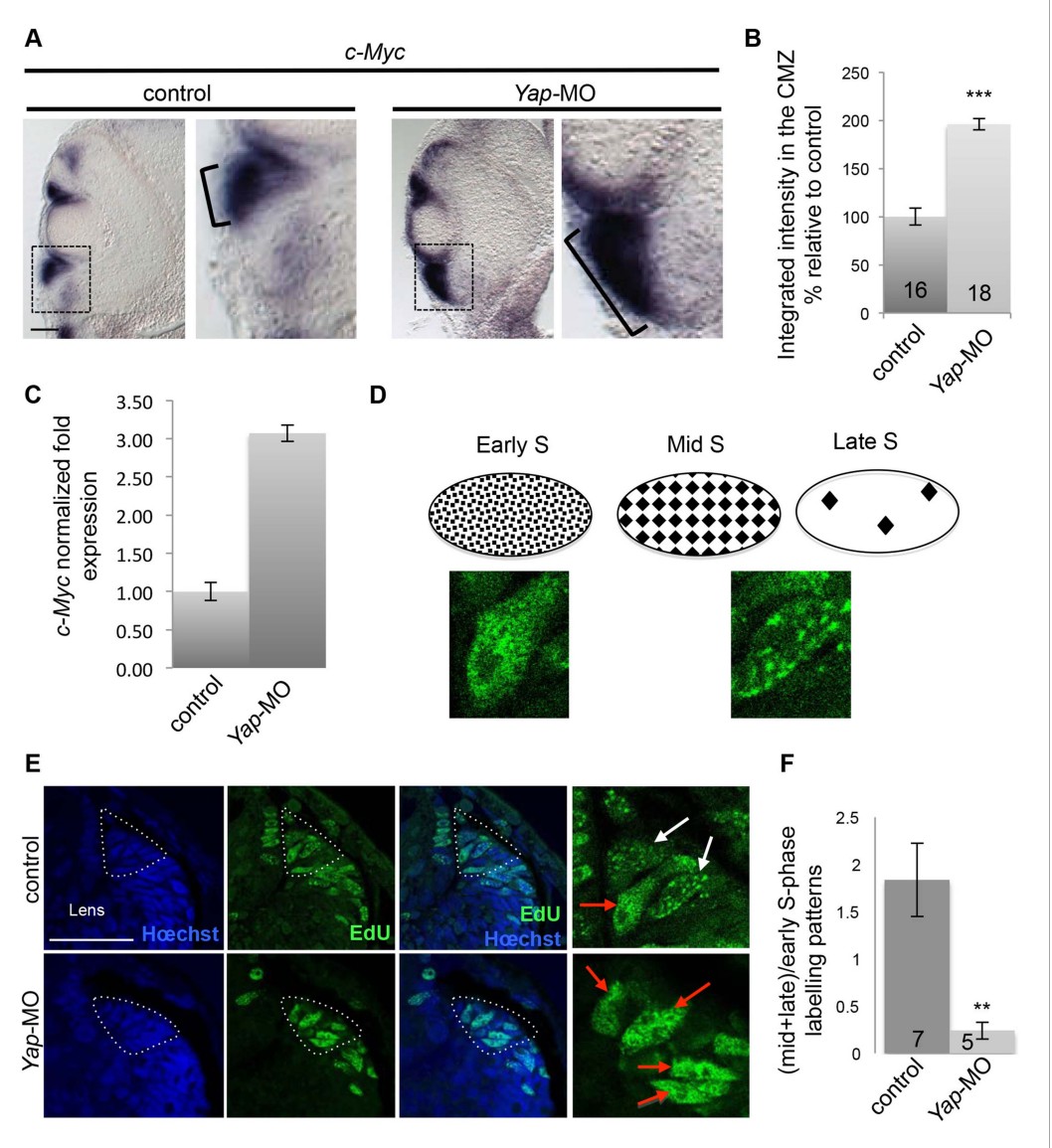

**Figure 5**. *Yap* loss of function affects the temporal program of retinal stem cell DNA replication. (**A**) In situ hybridization analysis of *c-Myc* expression on stage 40 retinal sections following two-cell stage microinjection of *Yap*-5-mismatch-MO (control) or *Yap*-MO. Images on the right show higher magnifications of the CMZ (dotted lines). Note the strong expansion of *c-Myc* expression area (bracket). (**B**) Quantification of the staining in the CMZ. The number of analyzed retinal sections per condition is indicated in each bar. (**C**) qPCR analysis of *c-Myc* expression in the retina of tadpoles injected as in (**A**). (**D**) Schematic representation of replication foci observed during S-phase progression, as inferred from EdU labeling. Pictures illustrate two examples of EdU-labeled foci observed in control CMZ cells, one homogenous (early S-phase) and one with large dots (mid/late S-phase). (**E**) Analysis of EdU-labeled replication foci (1 hr-pulse) in the CMZ of stage 40 tadpoles injected as in (**A**). Enlargements of the CMZ tip (dotted lines) are shown on the right. Early (red arrows) and mid/late profiles (white arrows) were distinguished.
(**F**) Corresponding quantification. The number of analyzed retinas per condition is indicated in each bar. Data are represented as mean ± SEM. Scale bar = 40 μm.

contribute to the reduced eye size, proliferating CMZ cells were pulse-labeled and their progeny chased and counted in the three retinal layers (*Figure 7E*). As expected, we found that fewer neurons were generated in a 2-day period in *Yap* morphant retinas compared to control ones (*Figure 7F*).

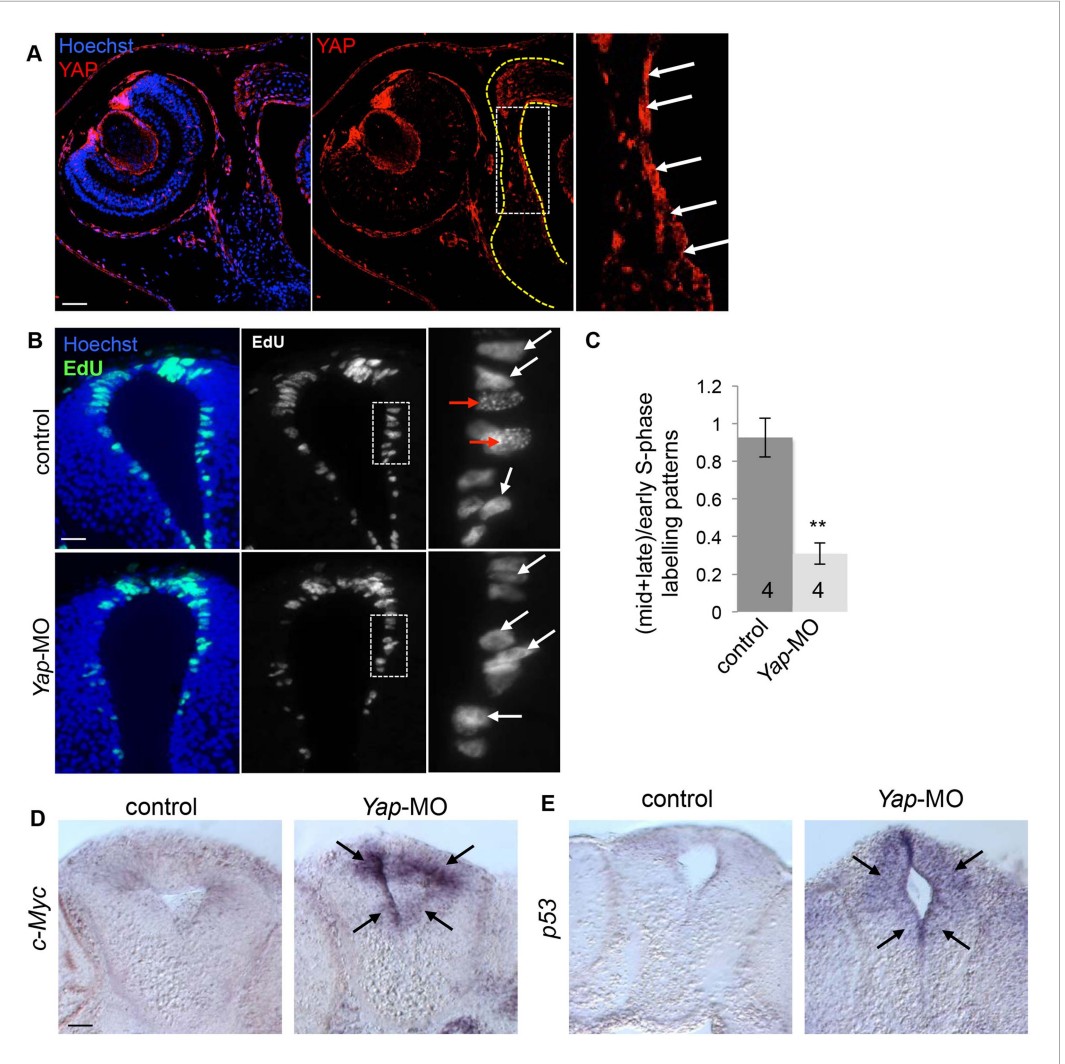

**Figure 6**. Effects of *Yap* knockdown in the neural tube. (**A**) Immunostaining with anti-YAP antibody on stage 42 sections. The left side of the neural tube is delineated with yellow dotted line. A higher magnification of the ventricular zone (white dotted line) is provided in the right panel. YAP labeling is most strongly detected in this region where progenitor cells reside (arrows). (**B**) Analysis of EdU-labeled replication foci (1 hr-pulse) in the neural tube of stage 40 tadpoles following two-cell stage microinjection of *Yap*-5-mismatch-MO (control) or *Yap*-MO. Enlargements (dotted lines) are shown on the right. Early (red arrows) and mid/late profiles (white arrows) were distinguished. (**C**) Corresponding quantification. The number of analyzed tadpoles per condition is indicated in each bar. Data are represented as mean ± SEM. (**D**, **E**) In situ hybridization analysis of *c-Myc* or *p53* expression in the neural tube of stage 40 tadpoles injected as in (**B**). Note the strong upregulation in the ventricular zone of the neural tube (black arrows). Scale bars = 40 μm.

## *Yap* knock-down in the CMZ activates the p53-p21 pathway

In order to gain additional insight into the molecular mechanisms underlying the *Yap* knockdown phenotype, we analyzed the expression of 15 genes encoding cell cycle regulators using the NanoString technology. Among them, only *p53* and *p21*^Cip1/WAF1 (previously named *Xic2* in *Xenopus* and *p21* hereafter) were significantly affected, with much higher levels present in *Yap* morphant retinas compared to control ones (*Figure 7G*). The tumor suppressor protein *p53* is activated in response to a variety of cellular stresses (including DNA damage) and triggers cell cycle arrest or apoptosis. In situ hybridization analysis revealed that its expression in the wild type retina is restricted to the CMZ. In addition and consistent with the NanoString data, *p53* staining was strongly enhanced in the CMZ of *Yap*-MO-injected tadpoles (*Figure 7H*). Of note, an up-regulation was

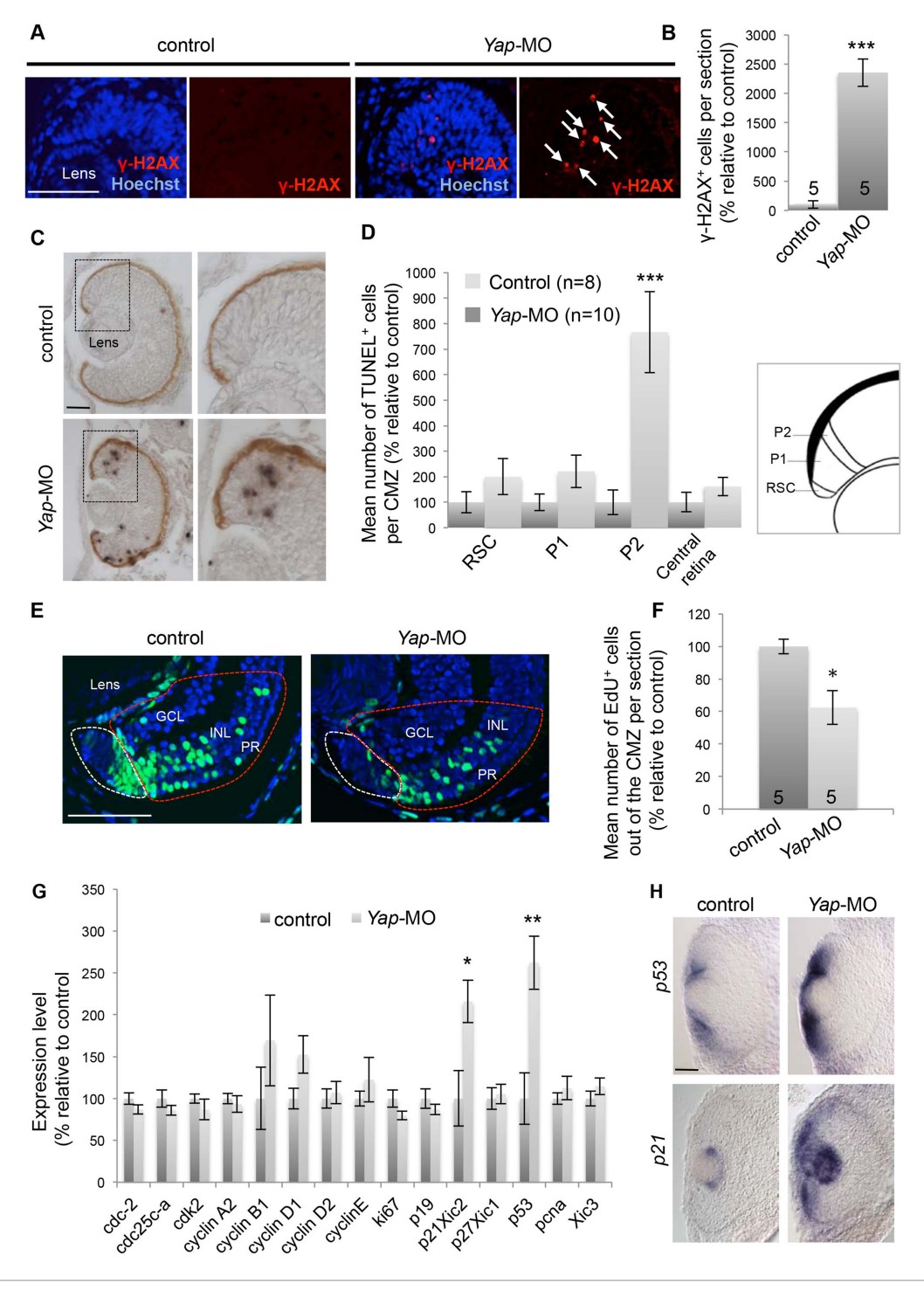

**Figure 7**. *Yap* loss of function induces DNA damage. (**A**) γ-H2AX immunolabeling in the CMZ of retinal sections from stage 40 tadpoles following two-cell stage microinjection of *Yap*-5-mismatch-MO (control) or *Yap*-MO. Arrows point to γ-H2AX-positive cells. (**B**) Corresponding quantification. (**C**) TUNEL assay on retinal sections from stage 40 tadpoles injected as in (**A**). Images on the right show higher magnifications of the CMZ delineated with dotted lines. (**D**) Quantification of TUNEL-positive cells in the different compartments of the CMZ as illustrated on the schematic. (**E**) 2 days-chase of EdU-labeled cells in the CMZ of stage 42 tadpoles injected as in (**A**). EdU-positive cells inside the zone encircled with a red dotted line have exited the CMZ (white dotted lines) and integrated the different retinal layers. GCL: ganglion cell layer; INL: inner nuclear layer; PR: photoreceptor layer. (**F**) Quantification of
*Figure 7. continued on next page*

Figure 7. Continued
EdU-positive cells in the neural retina layers. (**G**) mRNA expression levels of cell cycle genes as measured with the NanoString nCounter system in heads from stage 40 tadpoles following two-cell stage microinjection of Standard MO (control) or *Yap*-MO. Data are the mean of four independent experiments. (**H**) In situ hybridization analyses of *p53* and *p21* expression on retinal sections from stage 40 tadpoles injected as in (**A**). The number of analyzed retinas per condition is indicated in each bar (**B**, **F**) or on the graph (n in **D**). Data are represented as mean ± SEM. Scale bar = 40 μm.

observed in the neural tube as well (*Figure 6E*). p21 is a member of the CIP/KIP family of cyclin-dependent kinase inhibitors that blocks the G1-S transition and has emerged as a key player in the p53 pathway (*Attardi and DePinho, 2004*). Since it was previously described as a lens-specific marker in the *Xenopus* eye (*Daniels et al., 2004*), we asked if it could be linked with the observed CMZ phenotype. We thus performed in situ hybridization and observed a strong ectopic *p21* labeling within the CMZ of *Yap* morphant tadpoles (*Figure 7H*). As a whole, these results suggest that increased apoptosis and probably delayed cell cycle progression resulting from *Yap* knockdown might be driven by the p53-p21 pathway.

## PKNOX1 is a novel partner of YAP in the retina

As stated above, YAP is a co-transcriptional activator that functions in association with transcription factors such as its classical partners of the TEAD family. Among other interacting factors described so far is *Drosophila* Homothorax (*Peng et al., 2009*; *Zhang et al., 2011*). Interestingly, PKNOX1 (also named PREP1), a mammalian Homothorax ortholog belonging to the Meis/Prep homeodomain factor family, has recently been involved in the maintenance of genomic stability (*Iotti et al., 2011*). However, its physical interaction with YAP has not yet been reported in vertebrates. We found that the two proteins indeed interact in vitro using a two-hybrid assay (data not shown). In order to address whether they might do so in vivo as well, we performed a bimolecular fluorescence complementation (BiFC) experiment (*Figure 8A*) (*Ohashi and Mizuno, 2014*). In this purpose, constructs encoding YAP, PKNOX1 or TEAD1 (as a positive control for interaction with YAP) fused to either the amino or carboxyl-terminal fragment of the VENUS fluorescent protein were transfected in HEK293T cells (*Figure 8B* and data not shown for inverse VN/VC fusion combinations). As expected, co-transfection of *Yap* and *Tead1* constructs resulted in a significant nuclear BiFC signal. This was lost using a *Yap*$^{\Delta TBS}$ mutant devoid of TEAD binding site, validating the specificity of the BiFC staining. Co-transfection of both *Yap* and *pknox1* constructs led to a positive BiFC signal that mainly localized to the cytoplasm. YAP/PKNOX1 interaction was further confirmed by co-immunoprecipitation analyses following expression of tagged proteins (*Figure 8C*).

Since we found that *pknox1* is expressed in the CMZ (*Figure 9A*), we next sought for potential functional interaction with YAP in retinal stem cells. A loss of function approach was first undertaken to compare *Yap* and *pknox1* knockdown phenotypes (*Figure 9B–G*). The injected dose of *pknox1*-MO was adapted to avoid broad developmental defects (see *Figure 9—figure supplement 1* for validation of *pknox1*-MO specificity and efficiency). Although the eye phenotype appeared more dramatic than that observed upon *Yap* knockdown (layering defects of the retina), it similarly led to a significant reduction in total eye size (*Figure 9B,C*), associated with decreased EdU incorporation in the CMZ compared to controls (*Figure 9G,H*). In addition, analysis of EdU-labeled replication foci revealed a decreased proportion of stem cells in mid/late S-phase (*Figure 9D–E*) in *pknox1* morphant, as observed following *Yap* knockdown. These embryos also displayed upregulation of *c-Myc* expression in the CMZ, thus recapitulating main features of the *Yap* knockdown phenotype (*Figure 9F*). We then asked whether YAP and PKNOX1 might synergize in a co-overexpression assay. Of note, *pknox1* overexpression alone does not significantly affect CMZ cell proliferation. The injected dose of *Yap* mRNA was lowered so that it only leads to a moderate although significant increase in the number of EdU-labeled cells. In these conditions, *pknox1* mRNA injection was found to exacerbate *Yap* gain of function phenotype (*Figure 9I,J*). Finally, we tested whether *pknox1* knockdown might rescue the overproliferative effects of *Yap* misexpression. We found indeed that EdU incorporation was restored to a basal level in a *pknox1* morphant context (*Figure 9K,L*). Together, these data support the idea that PKNOX1 physically and functionally interacts with YAP in the CMZ.

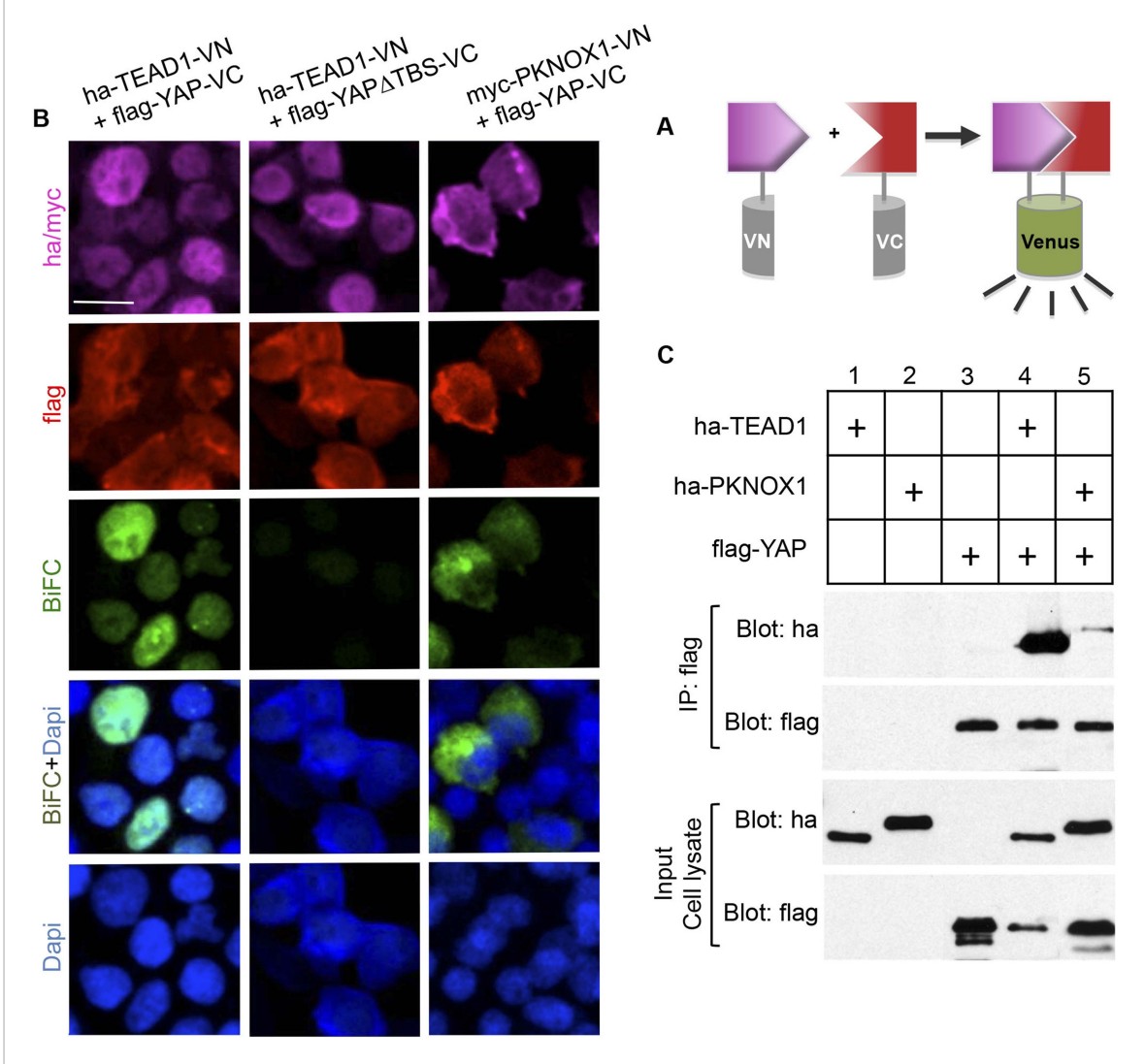

**Figure 8**. Physical interaction between YAP and PKNOX1. (**A**) Schematic representation of BiFC principle. (**B**) Immunolabeling/BiFC analysis on HEK293T cells transfected with VN and VC chimeric constructs, as indicated. (**C**) Co-immunoprecipitation assay of 293T cells transfected with tagged constructs as indicated. Scale bars = 20 µm.

## Discussion

Long-term maintenance of tissue homeostasis relies on the fine-tuning of adult stem cell activity. Our knowledge regarding the molecular basis sustaining somatic stemness features is still very limited but may have important implications for regenerative medicine and cancer therapy. In the present study, we identified YAP, a downstream effector of the Hippo pathway, as a stem cell specific marker required for homeostatic growth of the frog post-embryonic retina. Our in vivo loss of function approach unexpectedly revealed a novel role for YAP in governing DNA replication timing of retinal stem cells. We propose a model in which this function would contribute to the maintenance of their genomic stability (*Figure 10*). Importantly, based on our findings in the neural tube, we propose that such function might be generalizable to other neural precursor populations. Whether this involves YAP cytoplasmic and/or nuclear activity remains an open question.

The ability of nuclear YAP to expand stem/progenitor cell populations has been established in numerous model systems (*Barry and Camargo, 2013*; *Piccolo et al., 2014*). However, the mechanisms underlying altered cell proliferation (changes in cell cycle length and/or re-entry) are rarely investigated

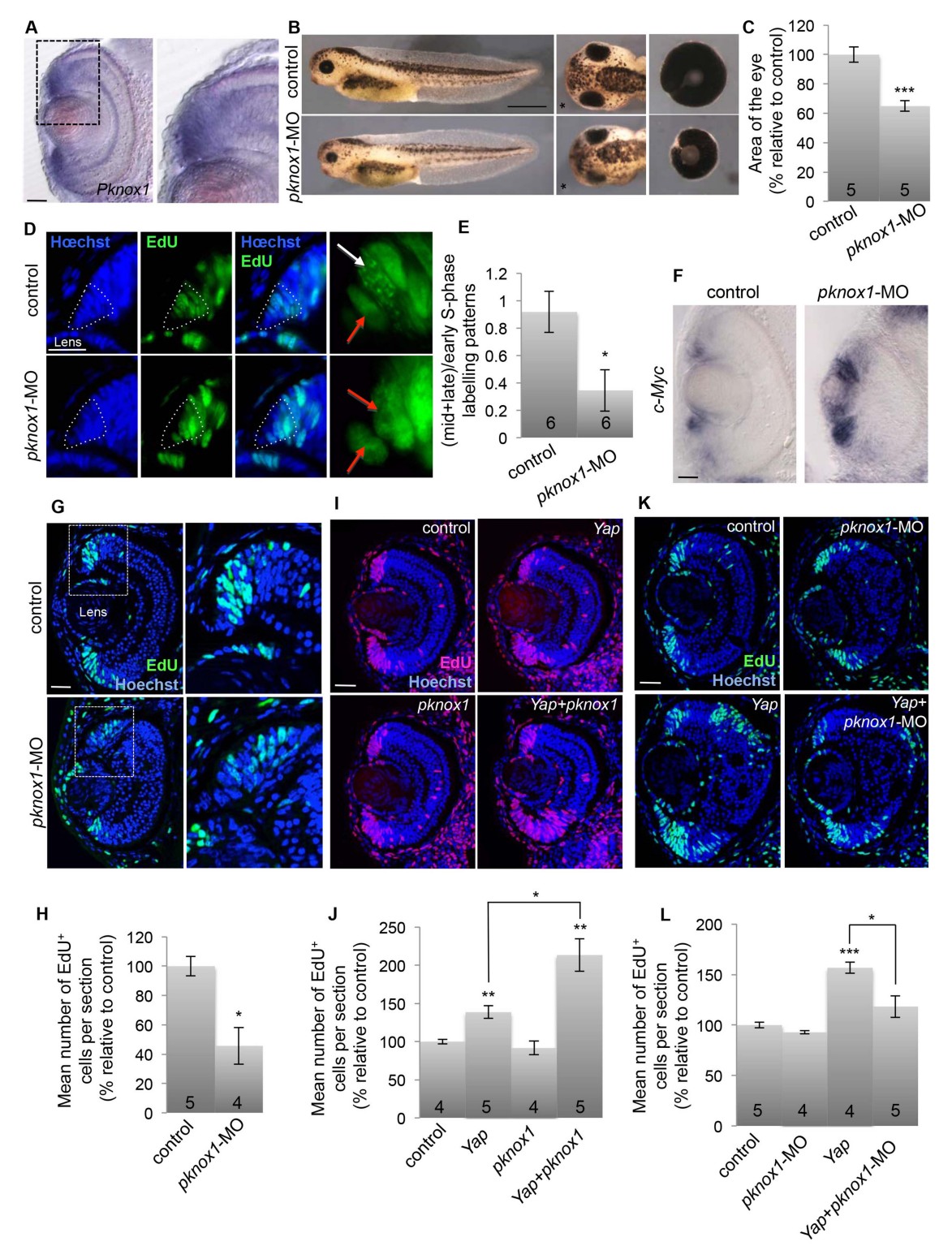

**Figure 9**. Functional interaction between YAP and PKNOX1. (**A**) In situ hybridization analysis of *pknox1* expression on stage 40 retinal sections. The right panels shows an enlargement of the CMZ region delineated with dotted lines. (**B**) Lateral views (left panels), head dorsal views (middle panels) and dissected eyes (right panels) of stage 40 tadpoles following two-cell stage microinjection of *pknox1*-5-mismatch-MO (control) or *pknox1*-MO. The asterisk indicates the injected side. (**C**) Quantification of dissected eye area. (**D**) Analysis of EdU-labeled replication foci (45 min-pulse) in the CMZ of tadpoles injected as in (**B**). Enlargements of the CMZ tip (dotted lines) are shown on the right. Early (red arrows) and mid/late profiles (white arrows)

*Figure 9. continued on next page*

*Figure 9. Continued*

were distinguished. (**E**) Corresponding quantification. (**F**) In situ hybridization analysis of *c-Myc* expression on stage 40 retinal sections from tadpoles injected as in (**B**). (**G–L**) EdU incorporation assays (3-hr pulse) analyzed on retinal sections from stage 40 tadpoles. (**G**, **H**) shows the effect of *pknox1* knockdown (injection of *pknox1*-5-mismatch-MO (control) or *pknox1*-MO). (**I**, **J**) shows the synergistic effects of *pknox1* and *Yap* (injection of *GFP* mRNA and either *ß-gal* mRNA (control), *Yap* + *ß-gal* mRNA (*Yap*), *pknox1* + *ß-gal* mRNA (*pknox1*), *Yap* + *pknox1* mRNA (*Yap* + *pknox1*)). (**K**, **L**) shows the rescue of *Yap* overexpression by *pknox1* knockdown (injection of either *pknox1*-5-mismatch-MO + *ß-gal* mRNA (control), *pknox1*-MO + *ß-gal* mRNA (*pknox1*-MO), *pknox1*-5-mismatch-MO + *Yap* mRNA (*Yap*), *pknox1*-MO + *Yap* mRNA (*Yap* + *pknox*-MO)). Of note, a suboptimal dose of *pknox1*-MO was used for the rescue experiment so that it does not alone give any eye phenotype. The total number of analyzed retinas per condition is indicated in each bar. Scale bar = 1 mm in (**B**) and 40 µm for all other panels.

The following figure supplement is available for figure 9:

**Figure supplement 1**. Validation of *pknox1*-MO efficiency and specificity.

in detail. In the developing neural tube for instance, YAP-driven increase in neural progenitor cell number has been proposed to result from accelerated cell cycle progression but whether its loss of function alters cell cycle kinetics as well remains an open question (*Cao et al., 2008*; *Zhang et al., 2012*). As observed in other adult organs in mammals (*Azzolin et al., 2014*; *Chen et al., 2014*; *Zhang et al., 2014*), our results suggest that YAP is dispensable for the maintenance of the stem cell pool. We instead uncovered that its depletion lengthens retinal stem cell divisions. However, to our surprise, this is associated with a dramatic shortening of their S-phase. It should be emphasized that such phenotype likely corresponds to a hypomorphic one due to the use of Morpholinos that allows partial loss of function. DNA replication is a tightly regulated process that follows a strict temporal program. Our observation that *Yap* knockdown results in a mark reduction of late S-phase labeling patterns can suggest that some firing origins have advanced their activation timing, which may account for the reduced S-phase duration. The genetic control of DNA replication temporal progression has not been elucidated yet and there are thus very few examples in the literature where gene perturbation leads to

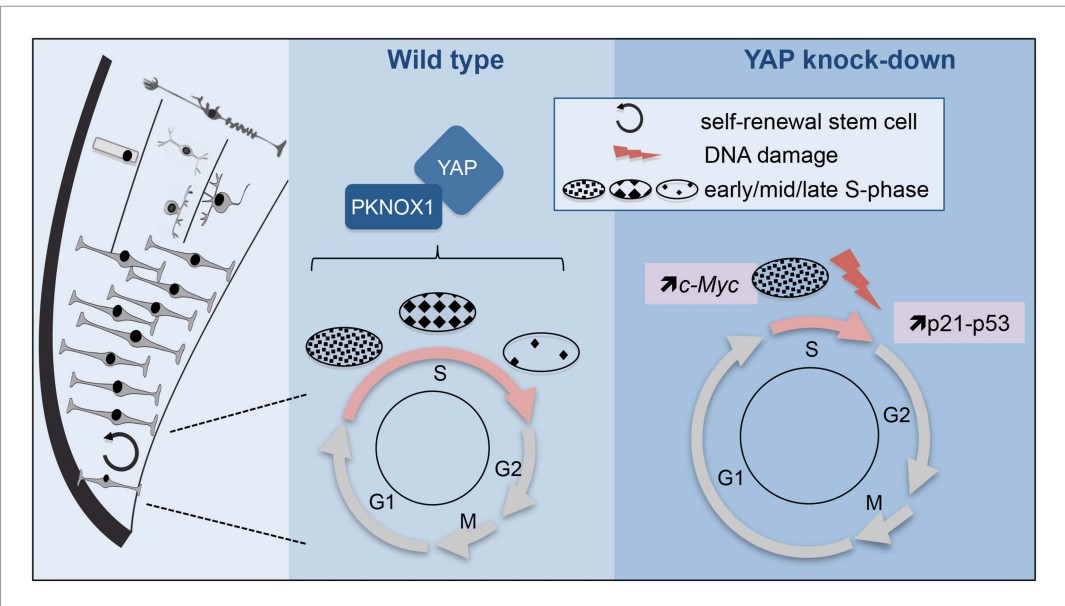

**Figure 10**. Model illustrating YAP function in retinal stem cells. We found that YAP is expressed in CMZ retinal stem cells (left panel). The middle panel shows the cell cycle of wild type retinal stem cells and the putative role of the YAP/PKNOX1 complex in the control of S-phase temporal progression (represented by the distinct patterns of DNA replication foci). YAP knock-down (right panel) leads to a dramatic reduction of S-phase length likely due to c-Myc-dependent premature firing of late replication origins. This results in increased occurrence of DNA damage, enhanced *p21* and *p53* expression and eventually cell death.

a deregulation of S-phase duration (*Aparicio, 2013*; *Yamazaki et al., 2013*). Beyond its well-characterized transcriptional activity, it was reported that c-Myc exerts a non-transcriptional control on DNA replication. Its overexpression indeed causes increased replication origin activity and consequent S-phase shortening (*Dominguez-Sola et al., 2007*; *Robinson et al., 2009*; *Srinivasan et al., 2013*). Intriguingly, in contrast to *Drosophila* or cancer cells, where *c-Myc* has been reported to be positively regulated by *Yap* (*Neto-Silva et al., 2010*; *Stocker, 2011*; *Xiao et al., 2013*), we found that its expression is enhanced in the CMZ of *Yap* morphant tadpoles. Although the underlying mechanism deserves further investigation, this raises the hypothesis that *c-Myc* may contribute to the S-phase defects caused by *Yap* knockdown. Besides, in addition to be involved in replication progression, we do not exclude that YAP may also regulate replication origin licencing in G1, as recently reported in human umbilical vein endothelial cells (*Shen and Stanger, 2015*). This might explain the observed lengthening of the cell cycle as a result of impaired G1/S transition. Alternatively, delayed G-phase progression might occur as a secondary consequence of S-phase defects.

Replication stress is a source of DNA damage, which may ultimately trigger activation of the p53-p21 pathway (*Bartek and Lukas, 2001*). As observed following *c-Myc* overexpression in vitro (*Dominguez-Sola et al., 2007*; *Robinson et al., 2009*; *Srinivasan et al., 2013*), we found an increased occurrence of double-strand breaks in *Yap* morphant retina, associated with an upregulation of both *p53* and *p21*. Since p21 is known to inhibit G1/S and G2/M transitions and p53 to induce programmed cell death (*Vogelstein et al., 2000*), this could contribute to both the lengthening in G phases and the increased number of apoptotic cells.

These findings raise key questions regarding specific features of stem cell biology. In addition to unique properties (such as the ability to self renew), emerging evidence suggests that somatic stem cells also differ from progenitor cells in the way they regulate basic cellular processes including their metabolic state (*Burgess et al., 2014*) or DNA-damage responses (*Insinga et al., 2014*). Regarding cell cycle progression, it has been shown during development that mammalian cortical neural stem cells exhibit a substantially longer S-phase than progenitors committed to neuron production (*Arai et al., 2011*; *Turrero García et al., 2015*). It was thus proposed that neural stem cells may need to invest more time during S-phase into quality control of replicated DNA. In agreement with this, we also found that CMZ retinal stem cells exhibit a longer S-phase compared to fast amplifying progenitors (data not shown). In addition, our work points for the first time towards a factor, YAP, that may be critically involved in this stem cell specific regulation of S-phase. Although its precise function in this process remains to be investigated, it indeed appears to be required for proper choreography of the DNA replication program and may as such be necessary to maintain genomic integrity of retinal stem cells.

Our findings also have important medical implications since aberrant DNA replication timing has been proposed to be a causative factor in diseases such as cancer and neuronal disorders (*Aladjem, 2007*; *Watanabe and Maekawa, 2010*; *Donley and Thayer, 2013*). Interestingly, another component of the Hippo pathway, LATS1, has very recently been involved in ATR-mediated response to replication stress in lung cancer cells (*Pefani et al., 2014*). Several Hippo pathway components may thus regulate (independently or in concert) S-phase progression and quality control and thereby safeguard genomic integrity.

Although *Homothorax* is known to partner the *Drosophila Yap* homologue *Yorkie* in some developmental contexts (*Peng et al., 2009*; *Zhang et al., 2011*), this has not been reported yet for its vertebrate orthologs. Here, we provide biochemical and functional evidences supporting an interaction between PKNOX1 and YAP in the retina. PKNOX1 belongs to the TALE (three amino acids loop extension) class of homeodomain proteins and is involved in many developmental processes (*Berthelsen et al., 1998*; *Ferretti et al., 2006*). Down-regulation of *pknox1* in both zebrafish and mouse embryos leads to a small eye phenotype (*Deflorian et al., 2004*; *Ferretti et al., 2006*), reminiscent of what we found in *Xenopus*. Interestingly, *pknox1* inactivation was reported to trigger cell death in the zebrafish CNS (*Deflorian et al., 2004*). Furthermore, *pknox1* deficiency leads to increased DNA damage and apoptosis both in embryonic fibroblasts and in the mouse epiblast (*Micali et al., 2009*; *Fernandez-Diaz et al., 2010*; *Iotti et al., 2011*). On the basis of these different reports and our findings, we thus propose that PKNOX1 and YAP interact together to maintain genomic stability in retinal stem cells. Whether PKNOX1 functions in competition with TEAD for YAP interaction or whether they all associate in a tripartite complex are important questions to be addressed in the future.

# Materials and methods

## Experimental procedures

### Ethics statement
All animal care and experimentation were conducted in accordance with institutional guidelines, under the institutional license C 91-471-102. The study protocol was approved by the institutional animal care committee: the Direction Départementale de la Protection des Populations.

### Plasmids and morpholinos
HA-tagged *Xenopus* constructs encoding wild-type or mutant YAP proteins (non-phosphorylable YAP$^{S98A}$ or TEAD-binding site-deleted YAP$^{\Delta TBS}$) were provided by S Gee and S Moody (*Gee et al., 2011*) and subcloned into pCS2+. The *Tead1* ORF, a gift from P Thiébaud (*Naye et al., 2007*), was subcloned into pCS2+Flag. The *pknox1* (*prep1*, NM_001096382) full-length cDNA sequence was cloned by RT-PCR into pCS2+Flag. *Yap*, *pknox1* and *Tead1* ORF were cloned in frame upstream the non-fluorescent N-ter or C-ter fragments of VENUS fluorescent protein (a gift from J Smith) in HA-, myc- or flag-tagged pCS2+ (*Saka et al., 2008*). PCR primer sequences are listed in *Supplementary file 1*. Translation-blocking antisense Morpholino oligonucleotides (MO, GeneTools, Philomath, OR, United States) and Photo-Morpholinos (Photo-MO, GeneTools) used in this study are also listed in *Supplementary file 1*.

### Microinjection
200 pg of mRNA (synthesized with mMessage mMachine kit, Life Technologies, Carlsbad, CA, United States) or 2 pmol MO were injected in one or two blastomeres at the two-cell stage. mRNAs encoding ß-Galactosidase or GFP were injected as controls and/or lineage tracers. In some experiments, the injected side was identified using the fluorescence of lissamine tagged-MO. Cleavage of antisense or sense Photo-MO was performed by exposing embryos/tadpoles to UV light (365 nm) for 10 min in a glass surrounded by aluminum. A ratio of 0.9:1 *Yap*-S-photo-MO to *Yap*-MO was used.

### EdU labeling, immunostaining and TUNEL assay
Tadpoles were injected intra-abdominally with 1 mM of 5-ethynyl-20-deoxyuridine (EdU, Invitrogen, Carlsbad, CA, United States) and fixed at the required stage in 4% paraformaldehyde. For cumulative labeling experiments, EdU was made constantly available for the desired time-period following injection by incubating tadpoles in a 1 mM EdU solution, renewed every other day. EdU incorporation was detected on paraffin sections using the Click-iT EdU Imaging Kit according to manufacturer's recommendations (Invitrogen). Immunostaining was performed following 4% paraformaldehyde fixation on paraffin sections with antibodies listed in *Supplementary file 2*. Standard procedures were used unless specified. Detection of apoptotic cells was carried out with the DeadEnd fluorometric TUNEL system (Promega, Fitchburg, WI, United States) or using TdT-driven dig-dUTP incorporation (Roche, Basel, Switzerland) followed by immunolabeling and NBT/BCIP staining, according to the manufacturer's instructions. Cell nuclei were stained with Hoechst (Sigma-Aldrich, St. Louis, MI, United States) or DAPI (Thermo scientific, Waltham, MA, United States).

### Analysis of cell-cycle parameters
For cumulative EdU incorporation assay in retinal stem cells, EdU-labeled and -unlabeled nuclei were scored among the 3 most peripheral CMZ cells on retinal sections (3–8 retinas analyzed per condition). The mean labeling index (LI) per retina was then plotted as a function of time after EdU injection. LI increases linearly until it reaches a plateau ($T_{plateau}$) allowing determining the growth fraction (GF; proportion of cycling cells within the considered population). The best-fit line was drawn using the Excel spreadsheet provided by R Nowakowski (*Nowakowski et al., 1989*). It allows estimating $T_c$ and $T_S$ using the two following equations: $T_{plateau} = T_c - T_s$; $LI_0 = GF \times (T_s/T_c)$ ($LI_0$ being the extrapolated y intercept of the best-fit line). Mitotic index and percentages of EdU-labeled mitosis were measured as previously described (*Locker et al., 2006*). The time required for half-maximal appearance of EdU labeling in the mitotic population was taken as an estimation of the mean G2 length ($T_{G2}$) (*Arai et al., 2011*).

### In situ hybridization
Digoxigenin-labeled antisense RNA probes were generated according to the manufacturer's instruction (Roche) from the following PBS plasmids: *cMyc*, a gift from WA Harris (*Xue and Harris,*

2011); *p21*, a gift from S Ohnuma (*Daniels et al., 2004*); *Yap*, a gift from S Moody (*Gee et al., 2011*); *Taz* (ImaGenes clone no. 102278); *Tead1* and *Tead2*, a gift from P Thiébaud (*Naye et al., 2007*). *p53* RNA probe was generated from a sequence encompassing the entire *p53* coding region that was amplified by RT-PCR and cloned into the pCS2+Flag vector (PCR primer sequences are listed in *Supplementary file 1*). Whole mount in situ hybridizations were carried out as previously described (*Perron, 2003*) and analyzed on 50 μm vibratome sections. For double EdU/*Yap* mRNA staining, in situ hybridization was performed on cryostat section as previously described (*Perron et al., 1998*) followed by EdU detection.

### RNA extraction
Total RNA from 50 tadpole heads (for subsequent NanoString experiment) or 70 dissected retinas (for subsequent qPCR) was isolated using the Trizol reagent (Life Technologies) and quality assessed using the Experion automated electrophoresis system (BioRad, Hercules, CA, United States).

### Quantitative real-time PCR
Reverse transcription was performed using the iScript cDNA Synthesis Kit (BioRad). qPCR reactions were performed in triplicate using SsoFast EvaGreen Supermix (BioRad) on a C1000 thermal cycler (CFX96 real-time system, BioRad). Results were normalized against the expression of reference genes ODC and RPL8 using CFX Manager software (BioRad). PCR primer sequences are listed in *Supplementary file 1*.

### Nanostring
The *nCounter Analysis* System (NanoString Technologies, Seattle, WA, United States) was used according to the manufacturer's instructions with 1000 ng of total RNA to profile 15 cell cycle genes for which we designed custom CodeSets (see *Supplementary file 3*). The readouts represent counts of individual fluorescent barcodes and provide a sensitive measurement of the selected RNA expression levels in the sample. The Nanostring data were analyzed using the nSolver Analysis Software 2. Background subtraction was performed by substracting the average plus two times the standard deviation of the 8 negative internal controls. Data were normalized as a geometric mean, against the internal positive control spikes and afterwards against three housekeeping genes (see *Supplementary file 3*).

### Co-immunoprecipitation assay and Western blot
Western blot were conducted using standard procedures on *Xenopus* embryo/tadpole protein extracts. Immunoprecipitation assays on HEK293T protein extracts were performed with the Dynabeads Protein A Immunoprecipitation Kit (Invitrogen) according to the manufacturer's protocol. Antibodies used are listed in *Supplementary file 2*.

### BiFC analysis
For BiFC experiments, plasmids (50 ng each) were transiently transfected in HEK293T cells using lipofectamine 2000 reagent (Invitrogen). Cells were grown for 16 hr, fixed and immunostained using standard procedures with antibodies listed in *Supplementary file 2*.

### Microscopy
Fluorescence and brightfield images were captured with an ApoTome-equiped Axio Imager.M2 microscope and processed using AxioVision REL 7.8 software (Zeiss, Oberkochen, Germany) or with a LSM 700 confocal microscope using ZEN software (Zeiss). Confocal pictures represent a merged z-stack of 15 slices (1 μm each).

### Quantification and statistical analyses
For quantifications of labeled cells by manual cell counting in the CMZ, 6 to 10 sections per retina and a minimum of 3 retinas were analyzed. Dissected eye area, Hoechst/PCNA labeling surface or in situ hybridization staining intensity in the CMZ were measured using Adobe Photoshop CS4 software. All experiments were performed at least in duplicate. Shown in figures are results from one representative experiment unless specified. Statistical analyses were performed by Student's *t*-test. Statistical significance is: *$p < 0.05$; **$p < 0.01$; ***$p < 0.001$; n.s. not significant.

## Acknowledgements
We thank S Gee, S Moody, P Thiébaud, WA Harris, S Ohuma and J Smith for providing plasmids. We are grateful to J Hamdache, E Braginskaja, A Ravigneaux for technical assistance and A Chesneau,

C de Medeiros and E Henry for animal care. We are indebted to K Marheineke for helpful discussions and grateful to X-J Yang for insightful comments on the manuscript. This research was supported by grants to MP from the ANR, Idex Paris-Saclay, Retina France, CNRS, Université Paris-Sud, and grants to KH from Cluster of Excellence and DFG Research Center Nanoscale Microscopy and Molecular Physiology of the Brain.

## Additional information

### Funding

| Funder | Grant reference | Author |
|---|---|---|
| L' Agence Nationale de la Recherche | ANR-10-BLAN-1220 | Muriel Perron |
| Campus Paris Saclay, Foundation de Cooperation Scientifique | Idex Paris-Saclay | Muriel Perron |
| Retina France | | Muriel Perron |
| Centre National de la Recherche Scientifique (National Center for Scientific Research) | | Muriel Perron |
| Univ Paris Sud | | Muriel Perron |
| Cluster of Excellence | Heidelberg University | Kristine A Henningfeld |

The funders had no role in study design, data collection and interpretation, or the decision to submit the work for publication.

### Author contributions
PC, GV-L, JB, ML, OB, MP, Conception and design, Acquisition of data, Analysis and interpretation of data, Drafting or revising the article; KP, RC, CM, CB, MH, KAH, Acquisition of data, Analysis and interpretation of data, Drafting or revising the article

### Author ORCIDs
Guillermo Vega-Lopez, http://orcid.org/0000-0002-2426-2844

### Ethics
Animal experimentation: All animal care and experimentation were conducted in accordance with institutional guidelines, under the institutional license C 91-471-102. The study protocol was approved by the institutional animal care committee: the Direction Départementale de la Protection des Populations.

## Additional files

### Supplementary files
• Supplementary file 1. Sequences of Morpholino oligonucleotides and primers used in the study.

• Supplementary file 2. List of antibodies used in the study.

• Supplementary file 3. Sequences used in the NanoString experiment.

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
