## [Decision Letter]

Thank you for submitting your work entitled “YAP controls retinal stem cell DNA replication timing and genomic stability” for peer review at *eLife*. Your submission has been favorably evaluated by Fiona Watt (Senior editor), a Reviewing editor, and three reviewers.

The reviewers have discussed the reviews with one another and the Reviewing editor has drafted this decision to help you prepare a revised submission.

All of the reviewers felt that this manuscript was very novel, interesting, and provided essential new insights into the function of YAP in neural stem cells in the *Xenopus* retina, and its function in cell cycle control and genomic stability. The authors showed that *Yap* loss results in elevated *c-Myc* expression, dramatically shortened S-phase with slower cell cycle progression, p53 induction and cell death. The accelerated S-phase is associated with a shift towards an “early” S-phase labeling patterns, increased DNA damage and death. Using morpholino knockdowns, fluorescence complementation analysis, overexpression and pull down assays they provide evidence for interaction of PKNOX1 with YAP. The reviewers were very impressed by the elegant approaches used, the high quality of the data throughout the manuscript, and the inclusion of many critical controls.

Issues to address in the revision:

A) Experimental

The reviewers felt that the study would be strengthened by some further characterization of the phenotype of overexpressed YAP. The knock down phenotype suggests that the cell cycle is extended by about 10hrs, which is the result of slowing down G2 by about 2h, shortening S-phase by about 13h, and assuming no change in M-phase, lengthening G1 by about 20h. Clearly the major affect is on G1 and S. It isn't very clear, why, if YAP-MO cells are rushing through S, why they also seem to hang up in G1. Perhaps cells that have made mistakes in the previous S have trouble in the G1-S transition. The reviewers felt there may be a clue to this in looking at what is happening in the overexpression/overactivity (overgrowth) phenotype. If YAP is affecting both of these phases of the cell cycle independently, then one would expect to see YAP overactivity both shorten G1 and lengthen S, but if the MO phenotype is primarily on S (inhibiting replication fork firing) then overactivity should slow down S but not affect G1 so much. The overactivity phenotype may also suppress *c-myc*, but this is problematic as *c-myc* is pro-proliferative. As there may be an antagonism *n-myc* and *c-myc* in this system, perhaps YAP-MO cells are biased to *c-myc* while YAP-overactive cells are biased to *n-myc*. A focused analysis on the cell cycle kinetics of the CMZ cells with a pulse of YAP should be considered.

While not essential, the reviewers also felt that it could strengthen the analysis if the authors determined if PNOX1 morphants also have elevated *c-Myc* expression.

B) Figure improvement and textual issues

1) When is YAP expressed in retinal development? Does it become gradually restricted to the CMZ? In Figure 1, it appears most *Yap* is cytoplasmic and not nuclear. The authors should comment on this. Do they think that the phenotypes caused by *yap* manipulation in the CMZ is due to cytoplasmic functions (and therefore interfering with other signaling pathways?).

2) The authors have done an excellent job in focussing in on stem cells and on the cell cycle effects, yet it is not altogether clear whether the cell cycle timing effects are particularly or only restricted to the stem cells. Are the YAP-controlled cell cycle dynamic effects quantifiably different in the stem cells vs. the progenitors?

3) In Figure 2, the *yap* morphant eye appears to indicate great cell packing and cell number, but the eye (at least this section) looks to be the same size as controls. The Authors should comment on this – particularly as the general conclusion is that *yap* knock-down reduces the cell cycle period, increases cell death and causes a small eye phenotype. Is this because the eyes in Figure 2 were taken at an earlier stage as compared to those described later on for the ‘final’ phenotype.

4) The formatting of Figure 3 is confusing. It would be improved by labeling the panels and images with the conditions rather than having the reader refer back to the table for explanation. (A clearer example is how similar data are represented in the supplemental data for Figure 3). More guidance in the Legend and/or text might help. Or perhaps more direct labels in C and D.

5) The authors should emphasize this is a knock-down/partial loss of function study. The use of morpholinos is particularly complicated as the per cell knock-down and dynamics of knock-down can vary tremendously. The authors should more fully acknowledge the caveats of morpholino knock-down and that the phenotypes likely relate to hypomophic *yap* activity, and not full loss.

6) Related to point 3, the authors should comment on the potential role of taz/wwtr1 in providing addition (compensatory/redundant) co-transcriptional activity to that of *Yap*. What is the status of Taz expression in the frog CMZ?

7) Last, although *Tead1* was used as a positive control for *yap* interaction, the function of the most-common *yap* nuclear interactant was not addressed. Is this in competition for *pknox1* binding, functioning in a tripartite complex, or simply not expressed in the CMZ? At minimum, given the canonical nature of *yap*/taz-tead activity, this should be commented upon.

---

## [Author Response]

Issues to address in the revision:

A) Experimental

*The reviewers felt that the study would be strengthened by some further characterization of the phenotype of overexpressed YAP. The knock down phenotype suggests that the cell cycle is extended by about 10hrs, which is the result of slowing down G2 by about 2h, shortening S-phase by about 13h, and assuming no change in M-phase, lengthening G1 by about 20h. Clearly the major affect is on G1 and S. It isn't very clear, why, if YAP-MO cells are rushing through S, why they also seem to hang up in G1. Perhaps cells that have made mistakes in the previous S have trouble in the G1-S transition. The reviewers felt there may be a clue to this in looking at what is happening in the overexpression/overactivity (overgrowth) phenotype. If YAP is affecting both of these phases of the cell cycle independently, then one would expect to see YAP overactivity both shorten G1 and lengthen S, but if the MO phenotype is primarily on S (inhibiting replication fork firing) then overactivity should slow down S but not affect G1 so much. The overactivity phenotype may also suppress* c-myc*, but this is problematic as* c-myc *is pro-proliferative. As there may be an antagonism* n-myc *and* c-myc *in this system, perhaps YAP-MO cells are biased to* c-myc *while YAP-overactive cells are biased to* n-myc*. A focused analysis on the cell cycle kinetics of the CMZ cells with a pulse of YAP should be considered.*

The question raised by the referee is very pertinent but extremely difficult to address in vivo. We do not have the tools to follow individual cells along their different rounds of division and can only measure global changes of cell cycle kinetics in a group of cells. As mentioned by the reviewer, this does not allow deciphering if G1 slow-down in *Yap* morphant CMZ is directly due to *Yap* depletion or a secondary consequence of S-phase defects. We agree that cell cycle kinetics analyses following *Yap* overexpression may provide some clues regarding that question. We thus performed two independent experiments but unfortunately, the analysis turned out to be more challenging than expected.

We first limited our analysis to the tip of the CMZ where stem cells reside. We however couldn’t find any difference in the EdU-labeling index between controls and *Yap*-overexpressing stem cells. We do not know whether this reflects that (i) YAP quantity is not limiting in these cells, (ii) whether YAP is not sufficient per se to slow down S-phase of stem cells (which already have a long S-phase) or (iii) whether homeostatic mechanisms maintain physiological levels of YAP activity in these *Yap*- expressing cells, as recently described in the mouse (Chen et al. Genes and Dev 2015 29(12):1285-97).

We thus wondered whether the impact of YAP on cell cycle kinetics could be detected in cells that normally do not express *Yap*. We therefore focused our analysis on CMZ progenitor cells. However, delimitation of the CMZ in many tadpoles was rendered extremely difficult by the deformations caused by the overgrowth phenotype (retinal folding). Consequently, we couldn’t discriminate post-mitotic neurons engulfed in the overgrown CMZ from EdU-negative proliferative cells. Our countings in CMZ progenitors are thus not trustable. Overall, we thus decided not to include these data in the manuscript. We thus leave open the question and accordingly modified the Discussion section: “Besides, in addition to be involved in replication progression, we do not exclude that YAP may also regulate replication origin licencing in G1, as recently reported in human umbilical vein endothelial cells (48) . This might explain the observed lengthening of the cell cycle as a result of impaired G1/S transition. Alternatively, delayed G-phase progression might occur as a secondary consequence of S-phase defects.”

*While not essential, the reviewers also felt that it could strengthen the analysis if the authors determined if PNOX1 morphants also have elevated* c-Myc *expression.*

As suggested, we performed *c-Myc* in situ hybridization on *pknox1* morphant retinas. We found that similarly to the *Yap* knockdown, *c-Myc* is upregulated in the CMZ compartment. This data is now added to Figure 9 (panel F) and mentioned in the Results section, last paragraph.

B) Figure improvement and textual issues

*1) When is YAP expressed in retinal development? Does it become gradually restricted to the CMZ?*?

To address the question of *Yap* expression in retinal development, we now provide in Figure 1—figure supplement 1 in situ hybridization panels during embryonic retinogenesis. At optic vesicle stages, *Yap* is faintly expressed in the presumptive neural retina (NR) but strongly labels the presumptive retinal pigment epithelium (RPE) as well as the NR/RPE border, a region that has been proposed to contain cells dedicated to form the CMZ (20). It then indeed gets restricted to the CMZ at later stages of development. These additional data shown in Figure 1—figure supplement 1 are mentioned in the Results section, subsection “Yap is expressed in slow dividing stem cells of the post-embryonic retina”.

*In*
Figure 1*, it appears most* Yap *is cytoplasmic and not nuclear. The authors should comment on this. Do they think that the phenotypes caused by* yap *manipulation in the CMZ is due to cytoplasmic functions (and therefore interfering with other signaling pathways?).*

As rightly noticed, YAP appears mainly cytoplasmic in CMZ cells. Yet, confocal analyses revealed small amounts of YAP protein in their nuclei as well. We added this information in the Results section: “YAP protein was detected in stem cells located at the tip of the CMZ, mainly in the cytoplasm, although some signal could be observed as well in the nuclei of these cells”. Whether the morphant CMZ phenotype is due to the loss of either or both YAP cytoplasmic or nuclear function thus remains an open question. We have raised this issue in the Discussion section, first paragraph.

2) The authors have done an excellent job in focussing in on stem cells and on the cell cycle effects, yet it is not altogether clear whether the cell cycle timing effects are particularly or only restricted to the stem cells. Are the YAP-controlled cell cycle dynamic effects quantifiably different in the stem cells vs. the progenitors?

The EdU cumulative labeling data we provide concern only 3 selected cells located in the most peripheral region, that we confidently assume to be stem cells. It would be much more complicated to delineate and only consider the *Yap*-expressing progenitor pool. Establishing objective limits between *Yap*-expressing stem and progenitor cells would require two additional stainings: one to label the whole domain where *Yap* is expressed and another to identify the stem cell compartment (in situ hybridization against a specific marker) . We don’t have the tools so far to perform such multiple labeling in addition to EdU and PH3 staining.

However, we did measure S-phase and total cell cycle lengths of all CMZ cells (but the 3 aforementioned stem cells). We found a 25% reduction in S-phase length (compared to 79% decrease when only stem cells are taken into account). This may reflect that *Yap* knockdown effects are more pronounced in stem *versus* progenitor cells. However, since we measured S-phase in a heterogeneous cohort composed of both *Yap*-expressing and *Yap*-negative progenitors, we cannot confidently conclude on that point. We thus prefer not to mention this in the revised manuscript.

*3) In*
Figure 2*, the* yap *morphant eye appears to indicate great cell packing and cell number, but the eye (at least this section) looks to be the same size as controls. The Authors should comment on this – particularly as the general conclusion is that* yap *knock-down reduces the cell cycle period, increases cell death and causes a small eye phenotype. Is this because the eyes in*
Figure 2
*were taken at an earlier stage as compared to those described later on for the ‘final’ phenotype.*

The eyes in Figure 2 were taken at the same stage than those described later on in the manuscript. There is however some variability in retinal size both in controls and morphant embryos. We agree that the chosen sections are not representative of the expected phenotype. To better illustrate retinal size reduction, we now show another picture for the control retina that comes from the very same experiment.

*4) The formatting of*
Figure 3
*is confusing. It would be improved by labeling the panels and images with the conditions rather than having the reader refer back to the table for explanation. (A clearer example is how similar data are represented in the supplemental data for*
Figure 3*). More guidance in the Legend and/or text might help. Or perhaps more direct labels in C and D.*

We have changed the formatting of Figure 3. Each panel is now labeled so there is no need to refer back each time to the table. We believe this indeed makes the figure easier to read. We also accordingly modified Figure 3—figure supplement 1.

*5) The authors should emphasize this is a knock-down/partial loss of function study. The use of morpholinos is particularly complicated as the per cell knock-down and dynamics of knock-down can vary tremendously. The authors should more fully acknowledge the caveats of morpholino knock-down and that the phenotypes likely relate to hypomophic* yap *activity, and not full loss.*

We now point out in the Discussion section, second paragraph, that the use of Morpholinos leads to a partial loss of function and therefore that the observed phenotypes are hypomorphic.

*6) Related to point 3, the authors should comment on the potential role of taz/wwtr1 in providing addition (compensatory/redundant) co-transcriptional activity to that of* Yap*. What is the status of Taz expression in the frog CMZ?*

As in other systems, TAZ might indeed compensate for at least part of YAP function. We cannot exclude that it is the case in the CMZ. However, in contrast to *Yap*, *Taz* seems faintly expressed in the postembryonic retina as inferred from a very weak and diffuse in situ hybridization signal. This is now added in the Result section, first paragraph, and shown in Figure 1—figure supplement 1.

*7) Last, although* Tead1 *was used as a positive control for* yap *interaction, the function of the most-common* yap *nuclear interactant was not addressed. Is this in competition for* pknox1 *binding, functioning in a tripartite complex, or simply not expressed in the CMZ? At minimum, given the canonical nature of* yap*/taz-tead activity, this should be commented upon.*

We agree with the reviewer that some information should be provided about TEAD/YAP potential interaction within the CMZ. We thus have added an in situ hybridization showing that *Tead1* and *Tead2* are both expressed in retinal stem cells of the CMZ, and as such are potential partners of YAP (Figure 1—figure supplement 1, and subsection “Yap is expressed in slow dividing stem cells of the post-embryonic retina”).

We also added an experiment showing that, in contrast to wild type YAP, a truncated construct of YAP lacking the TEAD binding site (*Yap*
^*TBS*^ ) is unable to trigger enhanced proliferation in the CMZ, as a wild-type YAP construct does. This suggests that the overproliferative phenotype in the retina requires interaction with a TEAD protein (Figure 1—figure supplement 3, and subsection “Yap overexpression promotes post-embryonic eye overgrowth”).

Whether TEAD and PKOX1 function in competition for YAP binding or in a tripartite complex is certainly an important issue to be addressed in the future. We now comment upon this in the Discussion section, last paragraph.

Additional comments:

We noticed that we omitted to mention the plasmids used for in situ hybridization in the Methods section in the previous version of the manuscript. We now added this information in the subsection “In situ Hybridization”.

We added in the Discussion section (fourth paragraph) a novel reference supporting that neural stem cells have a longer S -phase than progenitors committed to neuron production: Turrero García, M., Chang, Y., Arai, Y., and Huttner, W.B. (2015). S-phase duration is the main target of cell cycle regulation in neural progenitors of developing ferret neocortex. J. Comp. Neurol. [Epub ahead of print].